# Looking for ESKAPE Bacteria: Occurrence and Phenotypic Antimicrobial Resistance Profiles in Wild Birds from Northern and Central Italy Sites

**DOI:** 10.3390/antibiotics14101025

**Published:** 2025-10-14

**Authors:** Guido Grilli, Maria Cristina Rapi, Laura Musa, Giacomo Di Giacinto, Fabrizio Passamonti, Stefano Raimondi, Oriana Cianca, Maria Pia Franciosini

**Affiliations:** 1Department of Veterinary Medicine and Animal Sciences, University of Milan, Via dell’Università 6, 26900 Lodi, Italy; maria.rapi@unimi.it (M.C.R.); laura.musa@unimi.it (L.M.); 2Department of Veterinary Science, University of Turin, Largo Paolo Braccini 2, 10095 Grugliasco, Italy; giacomo.digiacinto@unito.it; 3Departments of Veterinary Medicine, University of Perugia, Via San Costanzo 4, 06100 Perugia, Italy; fabrizio.passamonti@unipg.it (F.P.); oriana.cianca273@gmail.com (O.C.); maria.franciosini@unipg.it (M.P.F.); 4Wildlife Rescue Centres (CRAS)—“Bosco WWF di Vanzago”, Via delle 3 Campane, 20043 Vanzago, Italy; stefanoraimondi67@gmail.com

**Keywords:** ESKAPE bacteria, wild birds, antimicrobial-resistance, One Health, MDR, *Escherichia coli*

## Abstract

Background/Objectives: Antimicrobial resistance is a critical global health challenge. Among resistant pathogens, the group of bacteria collectively referred to as ESKAPE (*Enterococcus faecium*, *Staphylococcus aureus*, *Klebsiella pneumoniae*, *Acinetobacter baumannii*, *Pseudomonas aeruginosa*, and *Enterobacter* spp.) is of particular concern due to their ability to evade multiple classes of antimicrobials. This study aimed to investigate the occurrence and resistance patterns of ESKAPE bacteria in wild birds from Northern and Central Italy sites, and to assess the presence of other bacteria of public health relevance. Methods: Cloacal swabs were collected from 141 wild birds. Samples were processed on selective and differential media, and bacterial identification was performed using Matrix-Assisted Laser Desorption/Ionization Time-of-Flight Mass Spectrometry. Antimicrobial susceptibility was evaluated through Minimum Inhibitory Concentration assays and interpreted according to international guidelines. Results: Thirty-seven isolates belonging to the ESKAPE group were identified: *E. faecium* (*n* = 10), *K. pneumoniae* (*n* = 9), *P. aeruginosa* (*n* = 8), *Enterobacter* spp. (*n* = 7), *S. aureus* (*n* = 2), and *A. baumannii* (*n* = 1). Multidrug-resistant isolates were observed among *K. pneumoniae* and *Enterobacter hormaechei*. *Escherichia coli*, although not included in the ESKAPE group, was frequently detected and often co-isolated with clinically relevant bacteria, highlighting its potential role as a reservoir of resistance genes. Conclusions: Wild birds can harbor resistant bacteria of clinical importance, including multidrug-resistant ESKAPE species. Their presence in avian populations underscores the role of wildlife in the environmental dissemination of antimicrobial resistance, with implications for both animal and human health.

## 1. Introduction

Antimicrobial resistance (AMR) is currently one of the most significant public health challenges, with substantial economic and social implications [1,2,3]. The selective pressure resulting from the prolonged use of antimicrobials in both human and veterinary medicine has driven the emergence of multidrug-resistant (MDR) pathogens capable of evading multiple classes of antimicrobial agents [4,5,6]. Consequently, continued research into innovative therapeutic strategies to combat AMR is essential. In this scenario, the ESKAPE bacteria (*Enterococcus faecium*, *Staphylococcus aureus*, *Klebsiella pneumoniae*, *Acinetobacter baumannii*, *Pseudomonas aeruginosa*, and *Enterobacter* spp.) are a source of concern [7,8]. These bacteria are a leading cause of mortality among immunocompromised subjects, especially in hospital and healthcare settings, due to their resistance not only to first-line antibiotics, but also to critically important antimicrobials (CIAs) in human medicine [7,8,9]. ESKAPE are not confined to clinical environments; they have also been detected in soil, surface water, and wastewater [10]. Their persistence in different habitats and ability to engage in horizontal gene transfer (HGT) allow them to acquire resistance through interactions with other bacteria [11]. In this respect, wildlife may serve as a link in the transmission cycle of AMR bacteria among humans, domestic animals, and the environment [12,13].

Although wild animals are not typically exposed to antimicrobials directly, their extensive use in human and veterinary medicine indirectly has driven the emergence of antimicrobial-resistant bacteria (ARB) in free-ranging species [14,15,16]. As ARB are generally present in natural ecosystems due to anthropogenic contamination, wild species inhabiting urban, agricultural, or landfill areas, or frequently interacting with humans and domestic animals, are more likely to carry ARB than those from less disturbed habitats [14,15,17,18,19]. Based on this, it is essential to identify wildlife species whose behavioral, ecological, and feeding characteristics make them suitable “sentinels” for assessing the extent of AMR in the environment [14,20]. Given their broad geographic distribution and high mobility, birds are acknowledged as effective bioindicators [17,21,22,23], capable of reflecting the presence and environmental dissemination of ARB potentially pathogenic to humans, ESKAPE group included [14,24,25]. Nevertheless, wild avian species are rarely included in AMR monitoring programs. Numerous studies have reported a high prevalence of AMR bacteria in bird populations inhabiting areas close to human settlements and with intensive livestock production in different geographical contexts [14]; however, considering the ESKAPE group, most of the available epidemiological investigations on wild birds have focused on single ESKAPE species or genera, while comprehensive assessments of the entire group remain limited [25]. Recent research has examined *S. aureus* in wild birds, characterizing resistance and virulence determinants [26], and *Enterococcus* spp. (including *E. faecium*) through combined phenotypic and genomic approaches [27]. Similarly, different studies have investigated the occurrence and MDR resistance profile, as well as Extended Spectrum Beta-Lactamases (ESBLs) production in *K. pneumoniae* isolates from different bird species and categories [28,29]. More recently, Silva et al. [25] reviewed extensively the diffusion of ESKAPE in wildlife, including avifauna, in several countries, highlighting the presence of resistant strains circulating among animals, men and the environment. In Italy, only one study conducted in the South by Russo et al. [12] specifically investigated the occurrence and resistance profiles of all bacterial species belonging to the ESKAPE group in avifauna, reporting that 24.5% of 163 birds sampled carried ESKAPE bacteria with variable AMR patterns, including resistance towards CIAs. To the best of our knowledge, no comprehensive study has yet investigated the simultaneous occurrence of all ESKAPE members in wild birds in Northern and Central Italy.

Considering this background, this study, carried out through collaboration between the Department of Veterinary Medicine, University of Perugia (DVMP, Umbria (Central Italy), and the Department of Veterinary Medicine and Animal Science, University of Milan (DVMM), Lombardy (Northern Italy), aimed to investigate, although in preliminary manner, the occurrence and phenotypic AMR profiles of ESKAPE bacteria isolated from wild birds. In addition, the presence of other bacteria of emerging relevance for public health [30,31,32] was documented. The detection of ESKAPE bacteria in wildlife carries important implications for zoonotic transmission and potential risks to human health. Although most research has focused on livestock and domestic animals as sources of clinically relevant resistant bacteria, wildlife constitutes a less monitored but equally relevant reservoir [25]. By providing novel data on the presence and resistance patterns of ESKAPE bacteria in wild avian species from Northern and Central Italy sites, this study provides additional data to help fill existing knowledge gaps and offers further insights into the potential role of wild birds in the clinically relevant ARB and AMR ecology. Furthermore, it highlights the importance of including wildlife in One Health-based surveillance frameworks to enable early detection of emerging resistant and pathogenic bacteria and to identify environmental hotspots before direct human exposure occurs [24].

## 2. Results

### 2.1. Sampled Bird Population

A total of 141 wild birds were sampled, with 66 (46.8%) from DVMM, and 75 (53.2%) from DVMP.

The investigated birds represented a broad taxonomic spectrum, including 15 orders, 22 families, and 42 species, comprising both resident and migratory species, as reported in Table 1. Overall, the most represented orders were *Passeriformes* (*n* = 45/141; 31.9%) and *Strigiformes* (*n* = 27/141; 19.1%), which together accounted for over 51% of all sampled birds. Other orders included *Columbiformes* (*n* = 16/141; 11.3%), *Apodiformes* (*n* = 12/141; 8.5%), *Falconiformes* (*n* = 10/141; 7.1%), *Accipitriformes* (*n* = 8/141; 5.7%), and *Anseriformes* (*n* = 8/141; 5.7%).

When comparing the two sites, differences emerged in species composition, as also illustrated in Table 1 and in Appendix A. DVMM had a higher representation of some *Anseriformes* species, particularly Mallard (*Anas platyrhynchos*), and *Apodiformes* species, represented by Common Swift (*Apus apus*) and Alpine Swift (*Tachymarptis melba*). On the other side, DVMP showed a greater number of *Strigiformes*, in particular Tawny Owl (*Strix aluco*). Similarly, diurnal raptors, *Accipitriformes* and *Falconiformes*, were more frequently recorded in Central Italy area.

### 2.2. Bacterial Isolation and Identification

#### 2.2.1. ESKAPE Bacteria Isolation in Both DVMM and DVMP

Out of 141 total cloacal swabs, 243 isolates were obtained and identified. Among these, 37 out of 243 (15.2%) were classified as ESKAPE bacteria, comprising 10 (4.1%) *E. faecium*, 9 (3.7%) *K. pneumoniae*, 8 (3.3%) *P. aeruginosa*, 7 (2.9%) *Enterobacter* spp., 2 (0.8%) *S. aureus*, and only 1 (0.4%) *A. baumannii*, as reported in Figure 1a. Of these, a total of 11 (29.7%) ESKAPE isolates were identified at the DVMP, while 26 (70%) were identified at the DVMM.

#### 2.2.2. Distribution of ESKAPE Bacteria and Their Co-Occurrence with Other Bacterial Isolates at DVMM

Among the ESKAPE isolates collected at DVMM, the most frequently identified were *E. faecium* (*n* = 9/26; 34.6%) and *K. pneumoniae* (*n* = 8/26; 30.8%), followed by *P. aeruginosa* (*n* = 5/26; 19.2%) and *Enterobacter* spp. (*n* = 3/26; 11.5%), as represented in Figure 1b. A single *A. baumannii* (3.8%) was isolated from a Eurasian Magpie (*Pica pica*). In several cases, ESKAPE bacteria isolated from DVMM samples were found in association with other relevant bacteria, as shown in Appendix A. Most *K. pneumoniae* isolates identified at DVMM were co-isolated with *Escherichia coli* and *Proteus mirabilis*, particularly in Common Swifts (*Apus apus*) and Common Blackbirds (*Turdus merula*). Two additional *K. pneumoniae* isolates were recovered alongside *E. faecium*: 1 from an Alpine Swift (*Tachymarptis melba*) and the other from a Common Swift (*Apus apus*). *Pseudomonas aeruginosa* was detected in association with *P. mirabilis* and *Klebsiella aerogenes* in a Common Blackbird (*Turdus merula*). Considering *Enterobacter* spp., notable co-isolations were observed, including *Enterobacter hormaechei* isolated with *Enterococcus faecalis* and *Staphylococcus saprophyticus* in a Little Egret (*Egretta garzetta*), and *Enterobacter roggenkampii* with *E. coli* in a European Greenfinch (*Chloris chloris*). Finally, the single *A. baumannii* isolate from DVMM was recovered together with *Klebsiella oxytoca* and *E. coli* in a Eurasian Magpie (*Pica Pica*).

#### 2.2.3. Distribution of ESKAPE Bacteria and Their Co-Occurrence with Other Bacterial Isolates at DVMP

Concerning DVMP samplings, among the 11 ESKAPE isolates obtained, the most frequently identified were *Enterobacter* spp. (*n* = 4/11; 36.4%), followed by *P. aeruginosa* (*n* = 3/11; 27.3%), *S. aureus* (*n* = 2/11; 18.2%), *K. pneumoniae* (*n* = 1/11; 9.1%), and *E. faecium* (*n* = 1/11; 9.1%), as reported in Figure 1b. Specifically, as shown in Appendix A, 4 *E. hormaechei* isolates were obtained: from a Mallard (*Anas platyrhynchos*) co-isolated with *P. aeruginosa*; from a Cormorant (*Phalacrocorax carbo*) with *E. coli*; from a European Starling (*Sturnus vulgaris*) with *Staphylococcus xylosus*; and from a Eurasian Hobby (*Falco subbuteo*) as a single isolate.

*Pseudomonas aeruginosa* was detected in a Eurasian Collared-Dove (*Streptopelia decaocto*), in a Short-toed Snake Eagle (*Circaetus gallicus*) alongside *Mammaliicoccus sciuri* and *E. faecium*, and in a Mallard (*Anas platyrhynchos*) in association with *E. hormaechei*. Of the 2 *S. aureus* isolates, 1 was recovered from a Common Blackbird (*Turdus merula*), while the other was identified in a Tawny Owl (*Strix aluco*) together with *S. xylosus* and *E. coli*. Additionally, the single isolates of *E. faecium* and *K. pneumoniae* were obtained from a Common Swift (*Apus apus*) and a Eurasian Goshawk (*Accipiter gentilis*), respectively (Appendix A).

#### 2.2.4. Non-ESKAPE Bacteria Isolation in Both DVMM and DVMP

As reported in Appendix A among non-ESKAPE bacteria, *E. coli* was the most prevalent species at both sampling sites, detected in 38 of 66 samples (57.6%) at DVMM and in 46 of 75 samples (61.3%) at DVMP. *Proteus mirabilis* was also frequently isolated in DVMM (*n* = 24/66; 36.4%), while *E. faecalis* was the most common Gram-positive bacterium (*n* = 33/66; 50%). Additional species of public health concern included *Citrobacter freundii*, *Citrobacter braakii*, and *Hafnia alvei*, the latter isolated from a Little Owl (*Athene noctua*) and from a Barn Swallow (*Hirundo rustica*). In DVMP *Escherichia marmotae* was also detected in several bird species. The 2 isolates of *Salmonella* Typhimurium were obtained from a Common Moorhen (*Gallinula chloropus*), and from a Blue Tit (*Cyanistes caeruleus*) associated with *S. saprophyticus*. Additionally, *K. oxytoca* and *K. aerogenes* were detected in both DVMP and DVMM (Appendix A).

### 2.3. Salmonella Serotyping

At the DVMP site, two *Salmonella* isolates were recovered: one from a Common Moorhen (*Gallinula chloropus*) and one from a Blue Tit (*Cyanistes caeruleus*). Both isolates were identified as *Salmonella enterica* subsp. *enterica* serovar Typhimurium, according to the Kaufmann–White–Le Minor scheme. No *Salmonella* isolates were obtained from DVMM samples.

### 2.4. AMR Profiles of ESKAPE Bacteria by Minimun Inhibitory Concentration (MIC) Determination

The AMR profiles of Gram-negative and Gram-positive ESKAPE bacteria isolated from DVMM and DVMP are summarized in Table 2 and Table 3, respectively.

#### 2.4.1. MIC Results for Gram-Negative ESKAPE Bacteria Isolated at DVMM

The AMR profile of Gram-negative ESKAPE isolates from DVMM are shown in Table 2. In details all *E. hormaechei* isolates were susceptible to the tested agents, whereas the single *E. roggenkampii* isolate, recovered from a European Greenfinch (*Chloris chloris*), was resistant to enrofloxacin, florfenicol, flumequine, sulfisoxazole, trimethoprim/sulfamethoxazole, and tetracycline. Among the *K. pneumoniae* isolates, resistance patterns varied: 4 out of 8 (50%) were fully susceptible to the tested antimicrobials, while the remaining isolates showed resistance to flumequine (*n* = 4/8; 50%), enrofloxacin (*n* = 4/8; 50%), cefazolin (*n* = 3/8; 37.5%), cefotaxime (*n* = 3/8; 37.5%), sulfisoxazole (*n* = 3/8; 37.5%), tetracycline (*n* = 3/8; 37.5%), amoxicillin/clavulanic acid (*n* = 2/8; 25%), and trimethoprim/sulfamethoxazole (*n* = 2/8; 25%). Resistance to aminosidine and gentamicin was also observed in some isolates. The single *A. baumannii* isolate, recovered from a Eurasian Magpie (*Pica pica*), was resistant to gentamicin but remained susceptible to trimethoprim/sulfamethoxazole, enrofloxacin and colistin. For several other antimicrobials, no validated clinical breakpoints are currently available for this species, and therefore MIC values could not be interpreted. *Pseudomonas aeruginosa* isolates were susceptible to colistin (*n* = 5/5; 100%), florfenicol (*n* = 5/5; 100%), and gentamicin (*n* = 5/5; 100%), while only 1 isolate (*n* = 1/5; 20%) resulted susceptible to enrofloxacin. Nonetheless, elevated MIC values for aminosidine and sulfisoxazole were observed in several isolates.

#### 2.4.2. MIC Results for Gram-Negative ESKAPE Bacteria Isolated at DVMP

With regard to the DVMP Gram-negative ESKAPE bacteria (Table 2), the single *K. pneumoniae* isolate, obtained from a Common Swift (*Apus apus*), was susceptible to all tested antimicrobials except cefazolin, for which intermediate susceptibility was observed. The *E. hormaechei* isolates exhibited high levels of resistance to enrofloxacin (*n* = 3/4; 75%), florfenicol (*n* = 3/4; 75%), sulfisoxazole (*n* = 3/4; 75%), flumequine (*n* = 3/4; 75%), trimethoprim/sulfamethoxazole (*n* = 3/4; 75%), and to third-generation cephalosporins (*n* = 2/4; 50%).

Similarly, to the DVMM site, *P. aeruginosa* isolates from DVMP exhibited elevated MIC values for sulfisoxazole (>512 µg/mL) and aminosidine (>32 µg/mL), yet remained susceptible to colistin, gentamicin, and florfenicol. Only 1 of the 3 (33.3%) isolated strain resulted resistant to enrofloxacin, while the other 2 presented reduced susceptibility to this molecule.

#### 2.4.3. MIC Results for Gram-Positive ESKAPE Bacteria Isolated at DVMM

Regarding Gram-positive ESKAPE bacteria from DVMM (Table 3), *E. faecium* isolates were uniformly susceptible to florfenicol (*n* = 9/9; 100%). Resistance was observed to enrofloxacin (*n* = 4/9; 44.4%), tetracycline (*n* = 3/9; 33.3%), erythromycin (*n* = 2/9; 22.2%), ampicillin (*n* = 1/9; 11.1%), amoxicillin/clavulanic acid (*n* = 1/9; 11.1%), and penicillin (*n* = 1/9; 11.1%).

#### 2.4.4. MIC Results for Gram-Positive ESKAPE Bacteria Isolated at DVMP

Among Gram-positive ESKAPE isolates from DVMP, both *S. aureus* isolates were resistant to sulfisoxazole and clindamycin, with 1 isolate also resistant to tetracycline. The single *E. faecium* isolate was resistant only to tetracycline, but susceptible to all other tested antimicrobials for which interpretation was feasible, as reported in Table 3.

### 2.5. MDR Profile Among ESKAPE Bacteria 

Of the 37 ESKAPE bacteria tested, 10 (27%) exhibited a MDR profile. Among these, 80% (*n* = 8/10) were Gram-negative, with 5 isolates from DVMM (62.5%) and 3 from DVMP (37.5%), while the remaining 20% (*n* = 2/10) were Gram-positive, one from each site.

#### 2.5.1. MDR in Gram-Negative ESKAPE Bacteria Isolated at Both Sites

Among the Gram-negative isolates from DVMM, MDR was observed in 5 cases: the single *E. roggenkampii* isolate and 4 out of 8 (50%) *Klebsiella pneumoniae* isolates recovered from a Common Redstart (*Phoenicurus phoenicurus*), European Pied Flycatcher (*Ficedula hypoleuca*), Alpine Swift (*Tachymarptis melba*), and Common Blackbird (*Turdus merula*). The MDR *K. pneumoniae* isolates were resistant to different combinations of beta-lactams, fluoroquinolones, tetracycline, and sulphonamides.

At DVMP, 3 of the 4 (75%) *E. hormaechei* isolates, obtained from a Mallard (*Anas platyrhynchos*), a Cormorant (*Phalacrocorax carbo*), and a Eurasian Hobby (*Falco subbuteo*), displayed MDR phenotypes, with resistance to third-generation cephalosporins, fluoroquinolones, phenicols, sulphonamides, and tetracyclines.

#### 2.5.2. MDR in Gram-Positive ESKAPE Bacteria Isolated at Both Sites

Regarding Gram-positive ESKAPE bacteria, MDR was less frequent. At DVMM, a single *E. faecium* isolate from a Common Woodpigeon (*Columba palumbus*) exhibited resistance to penicillins, macrolides, and tetracycline.

At DVMP, one *Staphylococcus aureus* isolate from a Common Blackbird (*Turdus merula*) showed resistance to lincosamides, sulphonamides, and tetracycline, thus fulfilling the MDR definition.

## 3. Discussion

Antimicrobial resistance is recognized as a serious public health threat and has been described as “*an ecological problem*” [35], involving complex interactions among different microbial populations that influence the health of animals, humans, and the environment [10]. The World Health Organization (WHO) has compiled a list of critical bacteria, identifying the ESKAPE bacteria as those for which urgent actions are required considering that, alongside *E. coli*, they are increasingly associated with MDR profile [36,37]. In this context, the present study aimed to provide preliminary insights into the potential role of wild birds in the dissemination of ESKAPE bacteria and to characterize their phenotypic AMR profiles.

In our study differences were observed among wild bird species sampled in the two sites: DVMM had a higher number of *Anseriformes* and *Apodiformes*, particularly Mallard (*Anas platyrhynchos*), Common Swift (*Apus apus*), and Alpine Swift (*Tachymarptis melba*) respectively, while DVMP showed a greater number of *Strigiformes*, such as Tawny Owl (*Strix aluco*) and Little Owl (*Athene noctua*), as well as diurnal raptors (*Accipitriformes*). This regional variation likely reflects differences in local habitat structure, landscape composition, and anthropogenic pressures between Lombardy’s peri-urban environment and Umbria’s more internal rural context, emphasizing the complementary ecological contributions of both sites to the study.

Among the ESKAPE bacteria, a total of 7 *Enterobacter* spp. isolates were recovered across both sampling sites. Six were identified as *E. hormaechei*, of which 4 were collected at DVMP (*n* = 4/75; 5.3%) and 2 at DVMM (*n* = 2/66; 3.0%), and 1 *E. roggenkampii* at DVMM (*n* = 1/66; 1.5%) from a European Greenfinch (*Chloris chloris*).

*Enterobacter* spp. are generally considered part of the intestinal microbiota of both humans and animals, where they are typically non-pathogenic [38,39]. Nonetheless, *E. hormaechei* and *E. roggenkampii* are members of the *Enterobacter cloacae* complex (ECC), a cluster of closely related species characterized by their potential to cause disease and their capacity to acquire and spread antimicrobial resistance genes (ARGs) [40,41]. *Enterobacter hormaechei*, in particular, is regarded as one of the most clinically significant species within the ECC, owing genomic islands associated with high pathogenicity [39]. Its public health importance has increased in recent years due to its capacity to disseminate ESBLs and carbapenemases [42], and the WHO has listed *E. hormaechei* among critically important pathogens [43,44,45,46,47]. In our study the co-occurrence of *E. hormaechei* with other relevant Gram-negative bacteria, such as *E. coli*, *P. aeruginosa*, and *K. pneumoniae* noticed in both DVMP and DVMM, is noteworthy. It is known that the plasmid-mediated AmpC beta-lactamase (AmpC) genes (e.g., *act-1*) in *E. hormaechei* can be transferred to other bacterial species, underscoring its potential to disseminate ARGs to bacteria of relevance in human medicine [48]. Regarding the AMR profiles of the isolates from DVMP, beyond their intrinsic resistance (IR) to ampicillin, amoxicillin/clavulanic acid, and cefazolin [33,34], *Enterobacter* spp. displayed acquired resistance to fluoroquinolones, quinolones, phenicols, and sulphonamides, with 75% (*n* = 3/4) showing a MDR profile. Similarly, MDR profile was observed in the single *E. roggenkampii* isolated at DVMM. By contrast, *E. hormaechei* isolates from DVMM were susceptible to all tested antimicrobials, possibly reflecting differences in environmental pressures or antimicrobial exposure [49]. Notably, 1 *E. hormaechei* isolated at DVMP isolated from a Common Starling (*Sturnus vulgaris*) was resistant to colistin, a last-resort antimicrobial for the treatment of MDR Gram-negative human infections [50,51,52]. These findings align with those of Russo et al. [12], who examined 163 wild birds in Southern Italy and detected *Enterobacteriaceae* in 53.9% of samples, including *Enterobacter* spp. These isolates frequently exhibited resistance to tetracycline, ciprofloxacin, and trimethoprim/sulfamethoxazole, closely mirroring the resistance profiles of the DVMP isolates in our study. Similarly, Foti et al. [53] reported the isolation of several *Enterobacter* species from wild birds, many of which were resistant to multiple antimicrobial classes, with high resistance rates to aminoglycosides and, to a lesser extent, fluoroquinolones. In addition, Gargano et al. [54] described *E. cloacae* carrying beta-lactam resistance determinants in migratory Woodcocks (*Scolopax rusticola*) from Italy. Although reports of *E. hormaechei* and *E. roggenkampii* isolated from wild birds are scarce, limiting the possibility of direct comparisons with existing studies, numerous investigations have described a high occurrence of MDR *Enterobacter* species in wild bird populations [12,53,55,56]. Together with our findings, this evidence reinforces the notion that wild birds may serve as reservoirs for MDR *Enterobacter* spp. across diverse geographic contexts.

*Pseudomonas aeruginosa*, widely distributed in the environment and frequently associated with both animals and humans [57] is recognized as a major nosocomial pathogen due to its intrinsic and acquired resistance mechanisms. For this reason, it has also been designated by the WHO as a priority target for the development of new antimicrobial therapies [36]. Resistance in *P. aeruginosa* is often linked to biofilm formation and the presence of mobile genetic elements encoding carbapenemases or ESBLs [58,59]. *Pseudomonas aeruginosa* was isolated from wild birds at both DVMP and DVMM sites. Specifically, this bacterium was recovered from 3 birds at DVMP and from 5 wild birds at DVMM, with a prevalence of 4% (*n* = 3/75) and 7.6% (*n* = 5/66), respectively. Aside from the IR of this species to beta-lactams and cephalosporins [33,34], the DVMM isolates were susceptible to most of the tested antimicrobials. Our findings are consistent with those of Russo et al. [12] who reported 18 (*n* = 18/163; 11%) *P. aeruginosa* isolates from various wild bird species, all of which were susceptible to the majority of the tested antimicrobials.

Conversely, other studies have reported markedly different resistance profiles among *P. aeruginosa* isolates from birds [60,61,62]. That said, it should be noted that for this species, the interpretation of susceptibility to several antimicrobials was limited by the absence of standardized clinical breakpoints in current reference guidelines. Nevertheless, certain considerations can be made regarding the MIC values obtained. In our isolates, MICs for flumequine, sulfisoxazole, and aminosidine could be considered high (>16 µg/mL, >512 µg/mL, and >32 µg/mL, respectively), with the exception of 1 DVMM isolate, recovered from a barn swallow (*Hirundo rustica*), which exhibited a MIC of 8 µg/mL for aminosidine.

*Klebsiella pneumoniae* is a Gram-negative bacillus included among the ESKAPE group due to its high propensity for acquiring and disseminating ARGs [36,63]. It is a well-recognized cause of serious healthcare-associated infections, including pneumonia, urinary tract infections, septicemia, and surgical site infections [64]. The increasing global prevalence of MDR and carbapenemase-producing *K. pneumoniae* isolates poses a significant public health threat, severely limiting therapeutic options and contributing to elevated mortality rates [64]. In the present study, a total of 9 *K. pneumoniae* isolates were recovered: 8 from wild birds sampled at DVMM and 1 from DVMP, corresponding to prevalence of 12.1% (*n* = 8/66) and 1.3% (*n* = 1/75), respectively. Excluding the IR to ampicillin reported by EUCAST [34] and CLSI [33] for this species, the DVMP isolate was intermediate to cefazolin but remained susceptible to all other tested antimicrobials. In contrast, the DVMM isolates demonstrated a concerning resistance pattern, with 50% (*n* = 4/8) classified as MDR. Resistance was most frequently observed to flumequine (*n* = 4/8; 50%), enrofloxacin (*n* = 4/8; 50%), tetracycline (*n* = 3/8; 37.5%), cefazolin (*n* = 3/8; 37.5%), cefotaxime (*n* = 3/8; 37.5%), and sulfisoxazole (*n* = 3/8; 37.5%). Additionally, 2 isolates (*n* = 2/8; 25%) were resistant to amoxicillin/clavulanic acid and to trimethoprim/sulfamethoxazole. These findings partially align with previous studies reporting a general trend of resistance to tetracyclines, sulfonamides, aminoglycosides, phenicols, and fluoroquinolones [65,66,67], and are also consistent with recent evidence indicating an increasing detection of *K. pneumoniae* resistant isolates in wildlife species, including birds [68]. Remarkably, all antimicrobial agents to which the *K. pneumoniae* isolates in this study exhibited resistance are classified as therapeutically important for human medicine [69].

*Acinetobacter baumannii* is a major opportunistic pathogen in healthcare settings and is also listed by the WHO as a “critical priority” for the development of new antimicrobials due to its frequent MDR phenotype [36,70]. In the present study, a single isolate was isolated from a Eurasian Magpie (*Pica pica*) at DVMM. These findings are consistent with previous studies on wild bird populations in different countries, where *A. baumannii* isolation rates have generally been low. For instance, Russo et al. [12] found no *A. baumannii* among 163 cloacal swabs from wild birds in Southern Italy, and similarly low detection rates have been reported in other geographical contexts [71]. On the opposite, the study by Dahiru et al. [72] in Nigeria reported *A. baumannii* in 31% (*n* = 15/48) of fecal samples from wild birds in agricultural settings. In view of its AMR profile, the single *A. baumannii* isolate, apart from IR displayed toward the most antimicrobial tested, was susceptible to colistin and trimethoprim/sulfamethoxazole, but resistant to gentamicin. Although only a single *A. baumannii* isolate was recovered in the present study, when considered alongside the existing literature this finding contributes to the evidence that wild avifauna may play a role, albeit limited, in the dissemination of *A. baumannii*, eventually including isolates of potential public health relevance [25,71,72].

Considering Gram-positive ESKAPE bacteria, *S. aureus* is a human pathogen associated with both community and hospital-acquired infections, ranging from mild skin inflammation to severe diseases such as pneumonia, sepsis, and endocarditis [73]. Its clinical relevance is further amplified by methicillin-resistant *S. aureus* (MRSA), which complicates treatment due to resistance to beta-lactams and other antimicrobial classes [73]. In our study, *S. aureus* isolates were detected only in DVMM, from a Common Blackbird (*Turdus merula*) and a Tawny Owl (*Strix aluco*), in association with *E. marmotae* and *E. coli*, respectively. Both isolates were resistant to sulfisoxazole and clindamycin but remained susceptible to most other antimicrobials, including fluoroquinolones, cephalosporins, and aminoglycosides. Sánchez-Ortiz et al. [74] reported that 82.6% (*n* = 19/23) of *Staphylococcus* spp. isolates from wounds in rehabilitated wild birds were resistant to at least one antimicrobial, with clindamycin resistance being common, while Carrel et al. [75] documented a progressive increase in resistance to sulphonamides (trimethoprim/sulfamethoxazole) in *S aureus* human isolates in USA. Comparatively, Russo et al. [12] isolated 6 *S. aureus* (*n* = 6/163; 3.7%) from wild birds, reporting more frequent resistance to CIAs than that observed in our study. However, the isolate from the Common Blackbird (*Turdus merula*) in DVMP also exhibited resistance to tetracycline, displaying an MDR phenotype. This finding is noteworthy in light of other studies reporting higher AMR rates in Gram-positive isolates from birds in rural and urban habitats [76,77,78,79]. Although the number of *S. aureus* isolates in the present study was limited, it is important to note that both isolates were phenotypically susceptible to oxacillin. While the CLSI now recommends cefoxitin as it provides more accurate detection of methicillin resistance phenotypes [33], oxacillin resistance testing has historically been considered a central step in the phenotypic characterization of staphylococci, as it was long regarded as the marker antimicrobial for detecting methicillin-resistant staphylococci (MRS) [80,81]. Based on these results, the *S. aureus* isolates from DVMP are unlikely to be classified as MRSA, although definitive conclusions cannot be drawn in the absence of further molecular investigations.

*Enterococcus faecium* is a gut commensal in humans and animals but is also can be a nosocomial pathogen [82]. Its clinical importance stems from its IR to several antimicrobial classes, including cephalosporins and aminoglycosides, and its increasing resistance to ampicillin and vancomycin [33,34,82,83]. For these reasons, *E. faecium* is among the WHO prioritized ESKAPE [36]. In our study, 10 *E. faecium* isolates were recovered, 9 (*n* = 9/66; 13.6%) from DVMM and 1 (*n* = 1/75; 1.3%) from DVMP. Across both sites, most isolates were susceptible to florfenicol and penicillin, whereas higher resistance rates were observed for tetracyclines, with the single isolate from DVMP and 3 of 9 isolates (33.3%) from DVMM being resistant. Resistance to enrofloxacin was also notable, detected in 44.4% (*n* = 4/9) of DVMM isolates, which were recovered from 3 specimens of Common Woodpigeon (*Columba palumbus*) and 1 from a Eurasian Hoopoe (*Upupa epops*), while the single DVMP isolate was intermediate to this molecule. Additionally, most of DVMM isolates exhibited reduced susceptibility to macrolides and rifampicin. This condition may suggest the presence of ARGs, such as *erm(B)*, although molecular characterization was beyond the scope of this study and warrants further investigation [82,84,85]. The findings of the present study are consistent with previous reports confirming the occurrence of resistance to fluoroquinolones, macrolides, and tetracyclines in *E. faecium* from wildlife [12,27,76,86]. Although resistance to these agents is so widespread that they are rarely used for the treatment of enterococcal infections [82], the results observed in this and similar studies are concerning, as they reveal substantial resistance to antimicrobials of public health relevance [58]. Notably, 1 *E. faecium* isolate from a Common Woodpigeon (*Columba palumbus*) at DVMM exhibited a MDR phenotype, showing resistance to penicillin, ampicillin, amoxicillin/clavulanic acid, enrofloxacin, erythromycin, and tetracycline. These results may underscore the influence of urban environments on the AMR, as the antimicrobials tested belong to the classes mostly used in humans as well as in companion animals, exerting considerable selective pressure at the environmental level [87,88]. Lastly, in the present study most *E. faecium* isolates exhibited uniformly elevated MIC values for kanamycin, with 2 isolates exceeding 250 µg/mL. This finding aligns with the well-documented IR of *E. faecium* to low-level concentration of several aminoglycosides, including kanamycin, mediated by a chromosomally encoded aminoglycoside-modifying enzyme [82,89]. Wild-type *E. faecium* typically shows kanamycin MICs in the range of 128–256 µg/mL; however, the markedly high MIC values (≥500 µg/mL) observed in the 2 DVM isolates may indicate the presence of additional acquired aminoglycoside resistance determinants [89], underscoring the need for further molecular characterization.

A special attention should be given to the high number of non-ESKAPE bacterial isolates (*n* = 206/243; 84.8%) recovered from both DVMM and DVMP, as several species are considered of relevance for public health.

*Escherichia coli* was the most frequently isolated bacterium, both at DVMM (*n* = 38/66; 57.6%) and DVMP (*n* = 46/75; 61.3%). Recent literature increasingly includes *E. coli* within the ESKAPE group, hence the term “ESKAPEE”, due to its frequent MDR phenotype and widespread environmental distribution [90,91]. In our samples, *E. coli* was often recovered in association with other bacteria, particularly Gram-positive bacteria such as *E. faecalis* and *S. aureus*, especially in passerines and *Columbiformes*. Such interspecies co-occurrence may enhance HGT, particularly in complex environmental microbiomes, thereby accelerating the spread of AMR among bacterial populations [92]. Additionally, 6 (*n* = 6/75; 8%) *E. marmotae* isolates were collected at DVMP, isolated with other relevant bacterial species, such as *S. aureus* and *E. faecalis.* This species is recognized as an emerging pathogen capable of acquiring ESBL and AmpC resistance genes [93]. Other non-ESKAPE but clinically significant bacteria detected included *Salmonella enterica* serovar Typhimurium and *Bacillus cereus*, both known not only as zoonotic agents, but also as bacteria with established resistance mechanisms [94]. These bacteria were recovered exclusively from DVMP with a prevalence of 2.7% (*n* = 2/75) and 1.3% (*n* = 1/75), respectively. *Citrobacter* spp., in particular *C. freundii*, *C. koseri*, and *C. braakii*, were isolated in both study sites. These opportunistic pathogens are known to harbor and exchange ARGs [32,95,96,97]. *Klebsiella oxytoca* and *K. aerogenes* were also detected, at DVMP from a Little Owl (*Athene noctua*) and a Barn Owl (*Tyto alba*), and at DVMM from a Eurasian Magpie (*Pica pica*) and a Common Blackbird (*Turdus merula*). These sedentary species share habitats with humans, potentially facilitating cross-species transmission of ARB. Although these bacteria are generally opportunistic pathogens in immunocompromised individuals, they can act as important vectors for the transfer of ARGs, including ESBLs and carbapenemases [98,99]. Finally, *Mammaliicoccus sciuri* (formerly *Staphylococcus sciuri*) and *S. xylosus*, common skin commensals with potential zoonotic relevance, were predominantly isolated in DVMP, particularly from raptors. *Mammalicoccus sciuri* is a recognized reservoir of *S. aureus mecA* homologues within Staphylococcal Cassette Chromosome (SCC*mec*) elements, owing to their high sequence homology and phylogenetic relatedness to *mecA* [100].

This study presents several strengths, including the simultaneous investigation of ESKAPE pathogens in wild birds from two geographically and ecologically distinct areas of Italy, Umbria and Lombardy. These regions were deliberately selected as they could represent markedly different ecological contexts, both in terms of natural environments and degrees of anthropogenic influence, thus providing a comprehensive overview of the occurrence of ESKAPE bacteria and their AMR profiles in wild birds. This dual-site approach can contribute a broader assessment of the potential role of avifauna in the dissemination of ARB across distinct Italian landscapes. Overall, the detection of AMR ESKAPE bacteria in both regional sites, particularly in Central Italy, where anthropogenic impact is lower compared to Lombardy, highlights the importance of wild birds as reservoirs and potential vectors of AMR bacteria and determinants, reinforcing the need for ongoing surveillance within a One Health framework. Furthermore, the inclusion of multiple avian species with diverse ecological niches increases the representativeness and ecological relevance of the sampled population. Nonetheless, some limitations should be acknowledged. First of all, the unequal number of samples collected between the two study sites, together with the biodiversity of the birds investigated may limit the robustness of direct statistical comparisons. Moreover, it should be emphasized that the primary aim of this work was descriptive in nature, focusing on the detection of ESKAPE bacteria in wild birds and the characterization of their AMR profiles. Accordingly, the study was not deliberately designed for inferential statistical analyses, which would have required larger and more homogeneous sample sizes. Rather, the value of the present investigation lies in providing baseline data that can contribute to addressing existing knowledge gaps and serve as a reference for future, more extensive studies. Another important limitation refers to the fact that, for several antimicrobial agents, the absence of established clinical breakpoints limited the interpretation of MIC values resulting in either under- or overestimation of resistance levels. The absence of molecular analyses, as this was not the objective of our study, restricted the ability to identify the genetic basis of the observed resistance phenotypes. To address this gap, targeted molecular analyses are planned as a follow-up to this work, thereby clarifying the genetic basis of the observed resistance phenotypes.

## 4. Materials and Methods

### 4.1. Sampling

A total of 141 cloacal swabs were collected from various species of wild birds between February and September 2024. In particular 75 samples were obtained from birds coming from the Wild Umbria Rescue Centre (Umbria region) or occasionally brought by private citizens and received at the DVMP hospital. The remaining 66 samples were obtained from birds housed at the WWF Wildlife Rescue Centre in Vanzago (Lombardy region), affiliated with the DVMM.

At the time of admission to the rehabilitation centers, as well as to the hospital at the DVMP site, an identification form was completed for each animal, recording the species, reason for admission, medical history, the date and location of finding.

Sampled birds were admitted due to various traumatic injuries, including those caused by collisions with vehicles, hunting or attacks by domestic animals. All cloacal swabs were collected, in both the rescue centers and/or, in the case, of DVMP, at the veterinary hospital, by the attending veterinarian during the clinical examination under standard, non-invasive conditions, before any antimicrobial treatment was given. After collection, cloacal swabs were placed in sterile tubes containing semisolid Amies transport medium (Copan Italia S.p.A., Brescia, Italy) and immediately stored at 4 °C and transported to the respective microbiology laboratories, either the Laboratory of Infectious Diseases in Perugia or the Avian Pathology Unit in Lodi (Milan), for further analyses.

### 4.2. Bacterial Isolation and Identification

To isolate and identify bacterial species from the sampled wild birds, an initial enrichment step was performed by inoculating each cloacal swab into sterile tubes containing 10 mL of Buffered Peptone Water (BPW) (ThermoFisher, Oxoid, UK), followed by incubation at 37 °C for 18–24 h. After enrichment, aliquots of the broth cultures were plated onto different growth media simultaneously, represented by Blood agar (ThermoFisher, Oxoid, UK), MacConkey agar (ThermoFisher, Oxoid, UK), and *Brilliance* UTI *Clarity* agar (ThermoFisher, Oxoid, UK), and incubated under aerobic conditions at 37 °C for an additional 18–24 h. Bacterial growth was subsequently examined, and representative colonies were selected based on morphological and phenotypic characteristics for species-level identification. This was achieved through Matrix-Assisted Laser Desorption/Ionization Time-of-Flight (MALDI-TOF) mass spectrometry (MS) using the MBT Microflex LT/SH system (Bruker Daltonik GmbH, Bremen, Germany), following the manufacturer’s instructions and protocol adapted from Rosa et al. [101], with the direct colony transfer method. Briefly, a small portion of each isolated colony was applied onto duplicate wells of disposable target plates using a sterile toothpick. The spots were then overlaid with 1 µL of α-cyano-4-hydroxycinnamic acid (HCCA) matrix solution (50% acetonitrile, 47.5% water, 2.5% trifluoroacetic acid; Bruker Daltonik GmbH, Bremen, Germany) and allowed to dry at room temperature. Mass spectra were then acquired in positive ion mode. Each run included duplicate spots of the Bruker Bacterial Test Standard. Spectra were analyzed against the MALDI Biotyper (MBT) Compass^®^ Library (HT) software (2023, Bruker Daltonik GmbH, Bremen, Germany), that encompasses 4320 species across 712 microorganism genera; bacterial identification was assigned based on the degree of overlap between the mass spectral profiles of the analyzed isolates and the reference spectra contained in the library. No subsequent serotyping analyses, nor phylogenetic or molecular confirmation methods, were applied in this study.

For the isolation of *Salmonella* spp., samples were processed according to the International Standard ISO 6579-1:2017 [102]. Colonies that growing on selective and differential media were confirmed and identified at the genus level using a MALDI-TOF mass spectrometer (MBT Microflex LT/SH, Bruker Daltonik GmbH, Bremen, Germany) applying the protocol described before. Serotyping was performed according to the Kaufmann–White–Le Minor scheme [103] using direct slide agglutination with the specific antisera (Statens Serum Institute, Copenhagen, Denmark).

### 4.3. Antimicrobial Susceptibility Testing (AST)

The antimicrobial susceptibility of ESKAPE isolates was assessed by broth microdilution method to determine the MIC value of each antimicrobial compound. All tests were performed at the Experimental Zooprophylactic Institute of Lombardy and Emilia-Romagna, “Bruno Ubertini”, Territorial Laboratory in Lodi. This laboratory provided full diagnostic support for both the DVMM and DVMP cohorts. Analyses were performed using the Sensititre System for MIC plate testing (Thermo Fisher Scientific, Oxoid, UK). To ensure the quality and accuracy of AST procedures, American Type Culture Collection (ATCC) standard reference (*E. coli* ATCC 25922, *S. aureus* ATCC 29213) were used. The testing procedures followed standardized protocols [104,105].

For Gram-negative ESKAPE isolates, the following antimicrobials were tested across defined concentration ranges: aminosidine (AN; 1–32 µg/mL), amoxicillin/clavulanic acid (AMC; 0.25–32 µg/mL), ampicillin (AMP; 0.25–32 µg/mL), cefazolin (CFZ; 0.5–8 µg/mL), cefotaxime (CTX; 0.5–4 µg/mL), colistin (CL; 0.03125–8 µg/mL), enrofloxacin (ENR; 0.015625–32 µg/mL), florfenicol (FFC; 1–64 µg/mL), flumequine (FLU; 1–16 µg/mL), gentamicin (GEN; 0.25–32 µg/mL), kanamycin (KAN; 2–32 µg/mL), sulfisoxazole (SFX; 128–512 µg/mL), tetracycline (TET; 0.5–16 µg/mL), and trimethoprim/sulfamethoxazole (SXT; 0.0625–16 µg/mL). For Gram-positive isolates, the antimicrobials tested included: amoxicillin/clavulanic acid (AMC; 0.25–16 µg/mL), ampicillin (AMP; 0.03125–16 µg/mL), cefazolin (CFZ; 0.25–8 µg/mL), ceftiofur (CTF; 0.25–8 µg/mL), clindamycin (DA; 0.5–2 µg/mL), enrofloxacin (ENR; 0.25–4 µg/mL), erythromycin (ERY; 0.03125–8 µg/mL), florfenicol (FFC; 2–8 µg/mL), kanamycin (KAN; 8–32 µg/mL and 250–500 µg/mL), oxacillin (OX; 0.25–4 µg/mL), penicillin (P; 0.03125–16 µg/mL), rifampicin (RD; 0.625–2 µg/mL), sulfisoxazole (SFX; 128–512 µg/mL), tetracycline (TET; 0.25–16 µg/mL), tilmicosin (TIL; 8–32 µg/mL), and trimethoprim/sulfamethoxazole (SXT; 0.125–8 µg/mL).

Bacterial susceptibility to each antimicrobial agent was first classified as susceptible (S), intermediate (I), and resistant (R) based on the breakpoints provided by the European Committee on Antimicrobial Susceptibility Testing (EUCAST), the Clinical and Laboratory Standards Institute (CLSI), or the Comité de l’Antibiogramme de la Société Française de Microbiologie (CA-SFM). Isolates showing intermediate susceptibility were then considered susceptible for the evaluation of the results [106,107]. As a general rule, EUCAST breakpoints [106,108] were preferentially applied. When not available, CLSI [33,109] and CA-SFM recommendations [110,111] breakpoints were used as an alternative reference. The latter were specifically used for the interpretation of MIC values for aminosidine (also known as paromomycin), as reported in other studies [112], and for kanamycin, particularly in *Enterobacteriaceae* and *S. aureus*, respectively. In general, preference was given to the most recent editions of the aforementioned guidelines; when breakpoints were not available in those versions, older editions were consulted. In the absence of established clinical breakpoints, epidemiological cut-off values (ECOFFs) defined for the target bacterial species, or, when unavailable, for closely related species, were considered [113]. Where no standardized breakpoints were available in any of the aforementioned sources, raw MIC values were reported. Specifically, given the absence of specific clinical breakpoints, MIC values for enrofloxacin were interpreted using those established for ciprofloxacin as a representative fluoroquinolone, consistently with the approach adopted in previous studies [114,115]. The sources used for the interpretation of AST results are reported in the Appendix A.

MDR bacteria were classified as those resistant to at least one antimicrobial agent in three or more distinct classes of antimicrobials [116].

## 5. Conclusions

This study investigated preliminarily the role of wild birds as potential reservoirs and sentinels of ARB, including clinically relevant ESKAPE bacteria. Among these, *Enterobacter* spp. were the most frequently isolated in DVMP, while *E. faecium* and *K. pneumoniae* were more commonly recovered in DVMM. Although *E. coli* is not universally included in the ESKAPE group, its frequent detection at both study sites, often in association with other Gram-positive and Gram-negative bacteria, highlights its ecological importance and potential role in HGT. Such interactions may facilitate the dissemination of AMR traits across bacterial species, enhancing microbial adaptability and persistence in different habitats.

Despite certain limitations related to our investigation, our results support the hypothesis that wild birds may harbor bacteria with both intrinsic and acquired resistance determinants. These include ESKAPE bacteria as well as other zoonotic or emerging pathogens of public health concern. The findings of this study reinforce the importance of integrating wildlife into AMR surveillance programs, particularly within a One Health framework, to better understand and mitigate the environmental dissemination of resistance. Continued monitoring in ecologically distinct areas, such as those represented by DVMM and DVMP, will be essential for identifying emerging threats and informing targeted interventions.

## Figures and Tables

**Figure 1 antibiotics-14-01025-f001:**
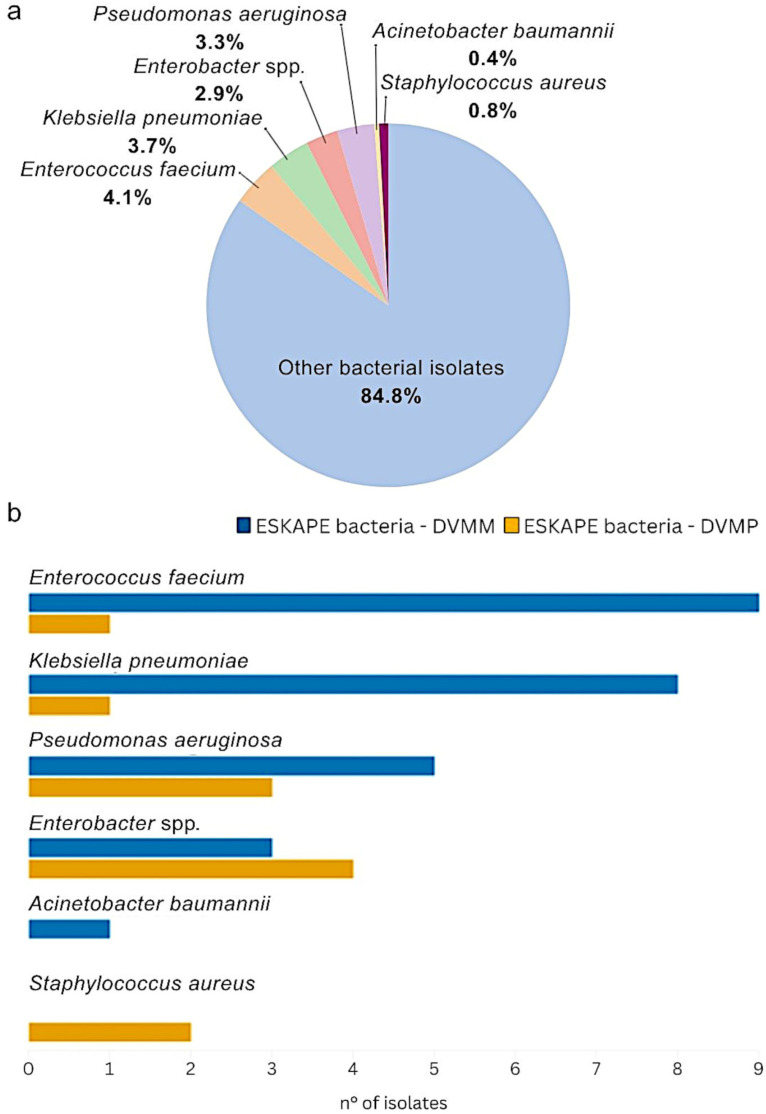
Bacterial isolates identified from wild birds. (**a**) Proportion of ESKAPE bacteria (15.2%) among the total isolates (*n* = 243), including *E. faecium* (4.1%), *K. pneumoniae* (3.7%), *P. aeruginosa* (3.3%), *Enterobacter* spp. (2.9%), *S. aureus* (0.8%), and *A. baumannii* (0.4%), from both DVMM and DVMP. (**b**) Number of ESKAPE isolates (*n* = 37) recovered in DVMM (*n* = 26) and DVMP (*n* = 11).

**Table 1 antibiotics-14-01025-t001:** Order, family, and species of wild birds and number of samples collected for each species at DVMM and DVMP.

Order	Family	Species	Samples DVMM	Samples DVMP	Total
*Accipitriformes*	*Accipitridae*	Eurasian Sparrowhawk (*Accipiter nisus* Linnaeus, 1758)	-	1	1
		Eurasian Common Buzzard (*Buteo buteo* Linnaeus, 1758)	-	3	3
		Goshawk (*Accipiter gentilis* Linnaeus, 1758)	-	1	1
		Short-toed Snake Eagle (*Circaetus gallicus* Gmelin, 1788)	-	3	3
*Anseriformes*	*Anatidae*	Mallard (*Anas platyrhynchos* Linnaeus, 1758)	6	1	7
		Swan (*Cignus olor* Gmelin, 1789)	-	1	1
*Apodiformes*	*Apodidae*	Alpine Swift (*Tachymarptis melba* Linnaeus, 1758)	3	-	3
		Common Swift (*Apus apus* Linnaeus, 1758)	8	1	9
*Bucerotiformes*	*Upupidae*	Eurasian Hoopoe (*Upupa epops* Linnaeus, 1758)	1	1	2
*Charadriiformes*	*Laridae*	Common Gull (*Larus canus* Linnaeus, 1758)	-	1	1
		European Herring Gull (*Larus argentatus* Pontoppidan, 1763)	-	1	1
*Columbiformes*	*Columbidae*	Common Woodpigeon (*Columba palumbus* Linnaeus, 1758)	5	-	5
		Eurasian Collared-dove (*Streptopelia decaocto* Frivaldszky, 1838)	5	3	8
		Common Pigeon (*Columba livia* Gmelin, 1789)	-	3	3
*Falconiformes*	*Falconidae*	Common Kestrel (*Falco tinnunculus* Linnaeus, 1758)	4	5	9
		Eurasian Hobby (*Falco subbuteo* Linnaeus, 1758)	-	1	1
*Galliformes*	*Phasianidae*	Pheasant (*Phasianus colchicus* Linnaeus, 1758)	-	1	1
*Gruiformes*	*Rallidae*	Common Moorhen (*Gallinula chloropus* Linnaeus, 1758)	-	1	1
*Passeriformes*	*Corvidae*	Hooded Crow (*Corvus cornix* Linnaeus, 1758)	4	2	6
		Eurasian Magpie (*Pica pica* Linnaeus, 1758)	5	3	8
		Eurasian Jay (*Garrulus glandarius* Linnaeus, 1758)	1	1	2
	*Turdidae*	Common Blackbird (*Turdus merula* Linnaeus, 1758)	7	5	12
	*Paridae*	Great Tit (*Parus major* Linnaeus, 1758)	1	-	1
		Blue Tit (*Cyanistes caeruleus* Linnaeus, 1758)	-	1	1
	*Muscicapidae*	Common Redstart (*Phoenicurus phoenicurus* Linnaeus, 1758)	1	1	2
		European Pied Flycatcher (*Ficedula hypoleuca* Pallas, 1764)	1	-	1
	*Fringillidae*	Eurasian Siskin (*Spinus spinus* Linnaeus, 1766)	-	1	1
		European Greenfinch (*Chloris chloris* Linnaeus, 1758)	4	1	5
	*Hirundinidae*	Barn Swallow (*Hirundo rustica* Linnaeus, 1758)	2	-	2
	*Sturnidae*	European Starling (*Sturnus vulgaris* Linnaeus, 1758)	-	4	4
*Pelecaniformes*	*Ardeidae*	Grey Heron (*Ardea cinerea* Linnaeus, 1758)	1	-	1
		Cattle Egret (*Bubulcus ibis* Linnaeus, 1758)	1	1	2
		Little Egret (*Egretta garzetta* Linnaeus, 1766)	1	-	1
		Night Heron (*Nycticorax nycticorax* Linnaeus, 1758)	-	1	1
*Piciformes*	*Picidae*	Green Woodpecker (*Picus viridis* Linnaeus, 1758)	-	1	1
*Podicipediformes*	*Podicipedidae*	Great Crested Grebe (*Podiceps cristatus* Linnaeus, 1758)	-	1	1
*Strigiformes*	*Strigidae*	Eurasian Scops-owl (*Otus scops* Linnaeus, 1758)	1	2	3
		Little Owl (*Athene noctua* Scopoli, 1769)	4	5	9
		Tawny Owl (*Strix aluco* Linnaeus, 1758)	-	9	9
		Owl (*Asio otus* Linnaeus, 1758)	-	3	3
	*Tytonidae*	Barn Owl (*Tyto alba* Scopoli, 1769)	-	3	3
*Suliformes*	*Phalacrocoracidae*	Cormorant (*Phalacrocorax carbo* Linnaeus, 1758)	-	2	2
	Total		66	75	141

**Table 2 antibiotics-14-01025-t002:** AMR profiles of Gram-negative ESKAPE isolates from wild birds at DVMM and DVMP sites by MIC determination.

			AST Panel													
			Aminoglycosides			Penicillins		Cephalosporins (1st–2nd)	Cephalosporins (3rd–4th)	Polymyxin	Fluoroquinolones	Phenicols	Quinolones	Sulphonamides		Tetracyclines
Site	Bird Species	ESKAPE Isolate	AN	GEN	KAN	AMP	AMC	CFZ	CTX	CL	ENR	FFC	FLU	SFX	SXT	TET
DVMM	Eurasian Magpie (*Pica pica*)	*Acinetobacter baumannii*	=2	R	≤2	IR	IR	IR	IR	S	S	=64	≤1	≤128	S	IR
	Eurasian Collared-dove (*Streptopelia decaocto*)	*Enterobacter hormaechei*	S	S	S	IR	IR	IR	S	S	S	S	S	S	S	S
	Little Egret (*Egretta garzetta*)	*Enterobacter hormaechei*	S	S	S	IR	IR	IR	S	S	S	S	S	S	S	S
	European Greenfinch (*Chloris chloris*)	*Enterobacter roggenkampii*	S	S	S	IR	IR	IR	S	S	R	R	R	R	R	R
	Common Blackbird (*Turdus merula*)	*Klebsiella pneumoniae*	S	S	S	IR	S	S	S	S	S	S	S	S	S	S
	Common Blackbird (*Turdus merula*)	*Klebsiella pneumoniae*	S	S	S	IR	R	S	S	S	R	S	R	S	S	R
	Common Redstart (*Phoenicurus phoenicurus*)	*Klebsiella pneumoniae*	S	S	S	IR	S	R	R	S	R	S	R	R	S	S
	European Pied Flycatcher (*Ficedula hypoleuca*)	*Klebsiella pneumoniae*	S	R	S	IR	R	R	R	S	R	S	R	R	R	R
	Common Swift (*Apus apus*)	*Klebsiella pneumoniae*	S	S	S	IR	S	S	S	S	S	S	S	S	S	S
	Alpine Swift (*Tachymarptis melba*)	*Klebsiella pneumoniae*	R	S	I	IR	S	R	R	S	R	S	R	R	R	R
	Alpine Swift (*Tachymarptis melba*)	*Klebsiella pneumoniae*	S	S	S	IR	S	S	S	S	S	S	S	S	S	S
	Eurasian Collared-dove (*Streptopelia decaocto*)	*Klebsiella pneumoniae*	S	S	S	IR	S	S	S	S	S	S	S	S	S	S
	Eurasian Collared-dove (*Streptopelia decaocto*)	*Pseudomonas aeruginosa*	>32	S	IR	IR	IR	IR	IR	S	S	S	>16	>512	IR	IR
	Great Tit (*Parus major*)	*Pseudomonas aeruginosa*	>32	S	IR	IR	IR	IR	IR	S	I	S	>16	>512	IR	IR
	Common Blackbird (*Turdus merula*)	*Pseudomonas aeruginosa*	>32	S	IR	IR	IR	IR	IR	S	I	S	>16	>512	IR	IR
	Barn Swallow (*Hirundo rustica*)	*Pseudomonas aeruginosa*	=8	S	IR	IR	IR	IR	IR	S	R	S	>16	>512	IR	IR
	European Greenfinch (*Chloris chloris*)	*Pseudomonas aeruginosa*	>32	S	IR	IR	IR	IR	IR	S	I	S	>16	>512	IR	IR
DVMP	European Starling (*Sturnus vulgaris*)	*Enterobacter hormaechei*	S	S	S	IR	IR	IR	S	R	S	S	S	S	S	S
	Cormorant (*Phalacrocorax carbo*)	*Enterobacter hormaechei*	S	S	S	IR	IR	IR	R	S	R	R	R	R	R	S
	Mallard (*Anas platyrhynchos*)	*Enterobacter hormaechei*	S	S	S	IR	IR	IR	R	S	R	R	R	R	R	I
	Eurasian Hobby (*Falco subbuteo*)	*Enterobacter hormaechei*	R	S	I	IR	IR	IR	S	S	R	R	R	R	R	R
	Common Swift (*Apus apus*)	*Klebsiella pneumoniae*	S	S	S	IR	S	I	S	S	S	S	S	S	S	S
	Short-toed Snake Eagle (*Circaetus gallicus*)	*Pseudomonas aeruginosa*	>32	S	IR	IR	IR	IR	IR	S	I	S	>16	>512	IR	IR
	Mallard (*Anas platyrhynchos*)	*Pseudomonas aeruginosa*	>32	S	IR	IR	IR	IR	IR	S	R	S	>16	>512	IR	IR
	Eurasian Collared-dove (*Streptopelia decaocto*)	*Pseudomonas aeruginosa*	>32	S	IR	IR	IR	IR	IR	S	I	S	>16	>512	IR	IR

S = susceptible (green color); I = intermediate (yellow color); R = resistant (red color); IR = intrinsic resistance (grey color) according to EUCAST and/or CLSI [33,34]. AN: aminosidine (1–32 µg/mL); AMC: amoxicillin/clavulanic acid (0.25–32 µg/mL); AMP: ampicillin (0.25–32 µg/mL); CFZ: cefazolin (0.5–8 µg/mL); CTX: cefotaxime (0.5–4 µg/mL); CL: colistin (0.03125–8 µg/mL); ENR: enrofloxacin (0.015625–32 µg/mL); FFC: florfenicol (1–64 µg/mL); FLU: flumequine (1–16 µg/mL); GEN: gentamicin (0.25–32 µg/mL); KAN: kanamycin (2–32 µg/mL); SFX: sulfisoxazole (128–512 µg/mL); TET: tetracycline (0.5–16 µg/mL); SXT: trimethoprim/sulfamethoxazole (0.0625–16 µg/mL).

**Table 3 antibiotics-14-01025-t003:** AMR profiles of Gram-positive ESKAPE isolates from wild birds at DVMM and DVMP sites by MIC determination.

			AST Panel															
			Penicillins				Cephalosporins (1st–2nd)	Cephalosporins (3rd–4th)	Lincosamides	Fluoroquinolones	Macrolides		Aminoglycosides	Phenicols	Ansamycins	Sulphonamides		Tetracyclines
Site	Bird Species	ESKAPE Isolate	OX	P	AMP	AMC	CFZ	CTF	DA	ENR	ERY	TIL	KAN	FFC	RD	SFX	SXT	TET
DVMM	Common Woodpigeon (*Columba palumbus*)	*Enterococcus faecium*	>4	S	S	S	IR	IR	IR	R	R	>32	≤250	S	I	IR	IR	S
	Eurasian Hoopoe (*Upupa epops*)	*Enterococcus faecium*	>4	S	S	S	IR	IR	IR	R	I	≤8	≤250	S	I	IR	IR	S
	Common Woodpigeon (*Columba palumbus*)	*Enterococcus faecium*	>4	S	S	S	IR	IR	IR	R	I	=16	≤250	S	I	IR	IR	S
	Common Swift (*Apus apus*)	*Enterococcus faecium*	>4	S	S	S	IR	IR	IR	I	I	=16	≤250	S	S	IR	IR	S
	Eurasian Jay (*Garrulus glandarius*)	*Enterococcus faecium*	>4	S	S	S	IR	IR	IR	S	I	=16	=500	S	I	IR	IR	R
	Common Swift (*Apus apus*)	*Enterococcus faecium*	>4	S	S	S	IR	IR	IR	I	I	=16	≤250	S	I	IR	IR	S
	Alpine Swift (*Tachymarptis melba*)	*Enterococcus faecium*	>4	S	S	S	IR	IR	IR	S	I	=16	≤250	S	S	IR	IR	S
	Common Woodpigeon (*Columba palumbus*)	*Enterococcus faecium*	>16	R	R	R	IR	IR	IR	R	R	>32	>500	S	I	IR	IR	R
	Eurasian Collared-dove (*Streptopelia decaocto*)	*Enterococcus faecium*	=1	S	S	S	IR	IR	IR	I	I	=16	≤250	S	S	IR	IR	R
DVMP	Eurasian Sparrowhawk (*Accipiter gentilis*)	*Enterococcus faecium*	=4	S	S	S	IR	IR	IR	I	S	=16	≤250	S	S	IR	IR	R
	Tawny Owl (*Strix aluco*)	*Staphylococcus aureus*	S	S	S	S	S	S	R	S	S	≤8	S	S	S	R	S	S
	Common Blackbird (*Turdus merula*)	*Staphylococcus aureus*	S	S	S	S	S	S	R	S	S	≤8	S	S	S	R	S	R

S = susceptible (green color); I = intermediate (yellow color); R = resistant (red color); IR = intrinsic resistance (grey color) according to CLSI and/or EUCAST [33,34]. OX: oxacillin (0.25–4 µg/mL); P: penicillin (0.03125–16 µg/mL); AMC: amoxicillin/clavulanic acid (0.25–16 µg/mL); AMP: ampicillin (0.03125–16 µg/mL); CFZ: cefazolin (0.25–8 µg/mL); CTF: ceftiofur (0.25–8 µg/mL); DA: clindamycin (0.5–2 µg/mL); ENR: enrofloxacin (0.25–4 µg/mL); ERY: erythromycin (0.03125–8 µg/mL);); TIL: tilmicosin (8–32 µg/mL); FFC: florfenicol (2–8 µg/mL); KAN: kanamycin (8–32 and 250–500 µg/mL); RD: rifampicin (0.625–2 µg/mL); SFX: sulfisoxazole (128–512 µg/mL); SXT: trimethoprim/sulfamethoxazole (0.125–8 µg/mL); TET: tetracycline (0.25–16 µg/mL).

## Data Availability

The original contributions presented in the study are included in the article; further inquiries can be directed to the corresponding author.

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
