# Peer review of "Looking for ESKAPE Bacteria: Occurrence and Phenotypic Antimicrobial Resistance Profiles in Wild Birds from Northern and Central Italy Sites"

_antibiotics, 2025, doi:10.3390/antibiotics14101025_

Round 1
Reviewer 1 Report
Comments and Suggestions for Authors
This study conducts a comprehensive investigation on the preliminarily the role of wild birds as potential reservoirs and sentinels of ARB, including clinically relevant ESKAPE bacteria. The findings are well supported by the results.
Here are some minor issues need further improvement:
In the introduction, it is recommended to emphasize more on the significance of this present study, and why yo choose wild birds as the objective.
In the methods, it is recommended to elaborate more about the bacteria species identification, and whether phylogenetic approaches were adopted?
The important parameters used MALDI TOF MS should be mentioned in the method.
It is recommend to cite more literature in the recent three years as references.
Author Response
|
Response to Reviewer 1 Comments
|
|
We sincerely would like to thank the reviewer for the time devoted to evaluating our manuscript and for the valuable comments provided. Please find the detailed responses below and the corresponding revisions in track changes in the re-submitted files. |
|
Comment and Suggestions |
|
1: In the introduction, it is recommended to emphasize more on the significance of this present study, and why you choose wild birds as the objective.
Response 1: We agree with the reviewer. As suggested, we have added several paragraphs to the Introduction (lines 97–115, 131–141) in order to provide a clearer understanding of the rationale behind the study and to better emphasize its significance within the current state of the art. |
|
2: In the methods, it is recommended to elaborate more about the bacteria species identification, and whether phylogenetic approaches were adopted?
Response 2: We would like to thank the reviewer for this recommendation. The Methods section has been expanded with further details on the MALDI-TOF MS procedure, in order to clarify how bacterial identification was performed in this study (lines 625-636). Related to this, no phylogenetic or molecular confirmation approaches were applied. Although the phylogenetic aspect can be an additional value, it was not the primary object in this study, chiefly focused on phenotypic characterization supported by MALDI-TOF MS identification of ESKAPE in wild birds in 2 different Italian sites. We have added a sentence in order to make this aspect clearer (line 637). |
|
3: The important parameters used MALDI TOF MS should be mentioned in the method.
Response 3: We appreciate the reviewer’s comment. Considering the previous suggestion, and subsequent answer, the Methods section has been revised to include the main parameters of the MALDI-TOF MS analysis, namely the instrument employed (MBT Microflex LT/SH, Bruker), the acquisition mode (positive ion), the matrix used (HCCA with detailed composition), the use of the Bacterial Test Standard, and the reference library applied (lines 625-636). |
|
4: It is recommended to cite more literature in the recent three years as references.
Response 4: We would like to thank the reviewer for this comment. As suggested, we have added some additional references, focusing on the 2022–2025 time frame (see references 26, 28, 29, 54, 62, 79). |
Reviewer 2 Report
Comments and Suggestions for Authors
Reviewer’s Comments and Suggestions for Authors
Journal: Antibiotics, MDPI
Manuscript ID: antibiotics-3876574
Type: Article
Title: Looking For ESKAPE Bacteria: Occurrence And Phenotypic Antimicrobial Resistance Profiles In Wild Birds From Northern And Central Italy Sites
Authors: Guido Grilli*, Maria Cristina Rapi, Laura Musa, Giacomo Di Giacinto, Fabrizio Passamonti, Stefano Raimondi, Oriana Cianca , Maria Pia Franciosini
The authors of the Manuscript ID: antibiotics-3876574 assessed the occurrence and antimicrobial resistance (AMR) phenotypic profiles of 37 ESKAPE bacteria isolated from 141 wild birds. The authors concluded that wild birds can harbor AMR ESKAPE isolates, including MDR isolates, as well as other bacteria relevant for public health.
The manuscript is presented too bad. Some methodological and analytical deficiencies that should be addressed.
Major essential revisions
- The Abstract is redundancy, and should be re-written.
- There shouldn’t be any abbreviations in the abstract. Abbreviations and acronyms are typically defined the first time the term is used within the abstract and again in the main text and then used throughout the remainder of the manuscript. Please consider adhering to this convention, and check throughout the manuscript.
- In the Introduction, the authors should describe some research background regarding occurrence and AMR phenotypes of ESKAPE bacteria isolated from wild birds in current literature.
- Line 52: please be careful of the word “microorganisms”, which includes bacteria, fungi, virus, etc. Please check the similar issue throughout the manuscript.
- In the Results, a more general summary at the endof each subsection should help improve readability and provide a clearer overview of the key findings.
- The Results section should be divided into more subtitles, and subtitles concerning the MDR, MIC values, and serotypes of the isolates should be added.
- Lines 91-108: this part should be included in a subtitle section.
- Lines 170-176: this part should be shifted to the Materials and Methods section.
- Lines 508-510: the authors mentioned that serotyping of Salmonella was performed. Please clarify related-results in the manuscript.
- In the 2.2 section, please rephrase the title of this section.
- Figure 1: the resolution of this figure is too low, and it should be increased. Also, the front size in this figure should be enlarged.
- Tables: the authors should format the tables as three-line tables. The similar issues in the tables in the Supplementary Materials.
- Table 1: the three columns on the right could be combined.
- Tables 2 and 3: these tables should be re-organized to be more easily read. I suggest that some of data in these tables could be presented in figures.
- In the Materials and Methods, how did the authors identify the diverse species of the wild birds?
- The experimental results lack statistical analysis, and doesn't specify the number of replicates or statistical tests used.
Author Response
|
Response to Reviewer 2 Comments
|
|
We would like to sincerely thank the reviewer for the time dedicated to the careful evaluation of our manuscript and, secondly, for the valuable suggestions that have guided revisions which we hope adequately address the concerns raised. Where it was not possible to implement specific modifications, we have provided detailed explanations in response to the reviewer’s comments. Please find the detailed responses below and the corresponding revisions in track changes in the re-submitted files. The line references provided in our responses correspond to the revised file. |
|
Major essential revisions |
|
1: The Abstract is redundancy, and should be re-written
|
|
Response 1: We thank the reviewer for this valuable observation. The Abstract has been carefully revised and restructured to remove redundancy and to improve clarity and conciseness (see lines 17–37). |
|
2: There shouldn’t be any abbreviations in the abstract. Abbreviations and acronyms are typically defined the first time the term is used within the abstract and again in the main text and then used throughout the remainder of the manuscript. Please consider adhering to this convention, and check throughout the manuscript.
|
|
Response 2: We sincerely thank the reviewer for this helpful observation. In accordance with the suggestion, all abbreviations and acronyms have been removed from the abstract, and the terms are now written in full (lines 17-37). Abbreviations are introduced only once in the main text, at their first occurrence, and then consistently used throughout the manuscript. We have also carefully re-checked the entire manuscript to ensure uniformity and compliance with this convention. |
|
3: In the Introduction, the authors should describe some research background regarding occurrence and AMR phenotypes of ESKAPE bacteria isolated from wild birds in current literature. Response 3: In accordance with reviewer’s appreciable comment we have expanded the Introduction (lines 97–115, 131–141) by adding recent literature on the occurrence and antimicrobial resistance phenotypes of ESKAPE bacteria isolated from wild birds. These additional literature has provided a stronger research background addressed to clarify the rationale of our research and better emphasize its significance within the current state of knowledge. |
|
4: Line 52: please be careful of the word “microorganisms”, which includes bacteria, fungi, virus, etc. Please check the similar issue throughout the manuscript. Response 4: We agree with the reviewer. The term “microorganisms” has been replaced with “bacteria” throughout the manuscript (lines 73, 126, 178, 348, 532, 536, 545) to ensure consistency with the focus of the study. The word “microorganisms” was retained only in line 634, as the MALDI Biotyper (MBT) Compass® reference library includes not only bacterial genera but also yeasts and filamentous fungi. |
|
5: In the Results, a more general summary at the end of each subsection should help improve readability and provide a clearer overview of the key findings. Response 5: We thank the reviewer for this suggestion. As recommended in the following comments, we have carefully reorganized the Results section by introducing additional subsections, providing a clearer structure and facilitating the reader’s understanding of the main findings. Hoping that this reorganization will be sufficient to improve readability, we preferred not to add further summarizing sentences at the end of each subsection, to avoid some redundancy with the detailed descriptions already provided. |
|
6: The Results section should be divided into more subtitles, and subtitles concerning the MDR, MIC values, and serotypes of the isolates should be added. |
|
7: Lines 91-108: this part should be included in a subtitle section. Response 7: We thank the reviewer for this helpful comment. As suggested, the indicated part has been placed under a new subtitle section to improve the structure and readability of the Introduction (lines 143-161). |
|
8: Lines 170-176: this part should be shifted to the Materials and Methods section. Response 8: We thank the reviewer for this observation. This part has been removed from the Results section (lines 242-249), to avoid avoiding redundancy and improving conciseness of the manuscript. |
|
9: Lines 508-510: the authors mentioned that serotyping of Salmonella was performed. Please clarify related-results in the manuscript. Response 9: We thank the reviewer for this comment. A dedicated subsection has been added to the Results section (lines 221-226), where the outcomes of Salmonella serotyping are now reported. |
|
10: In the 2.2 section, please rephrase the title of this section. Response 10: Thank you for your comment. The title of Section 2.2 has been rephrased accordingly to improve clarity (line 239). |
|
11: Figure 1: the resolution of this figure is too low, and it should be increased. Also, the front size in this figure should be enlarged Response 11: Thank you for the wright observation. Figure 1 has been replaced with a higher-resolution version, and the font size of all labels and legends has been increased to improve readability. In response to this comment, we considered it appropriate to also improve the resolution of Figure S1 in the Supplementary Materials. |
|
12: Tables: the authors should format the tables as three-line tables. The similar issues in the tables in the Supplementary Materials. Response 12: We thank the reviewer for this suggestion. We have reformatted the tables into the three-line style as much as possible, thereby improving consistency and presentation. In a few cases, however, some additional horizontal divisions were intentionally retained: in Table S3 (Supplementary Materials), to clearly separate different bacterial species, and in Tables 2 and Table 3, to distinguish between DVMM and DVMP isolates. These adjustments were made to preserve clarity and facilitate the interpretation of complex datasets, while still aligning the tables as closely as possible with the requested format. The Table 1 was left unchanged, as we considered its current structure more suitable for presenting the diversity of the sampled bird species in a clear and accessible way. |
|
13: Table 1: the three columns on the right could be combined. Response 13: We carefully considered the reviewer’s suggestion of merging the three rightmost columns of Table 1. However, we preferred to keep the current layout, as we believe it provides a clearer presentation of the data. In our opinion, merging the columns would risk reducing the readability of the table and make interpretation more complex for the reader. We hope the reviewer will understand our choice. |
|
14: Tables 2 and 3: these tables should be re-organized to be more easily read. I suggest that some of data in these tables could be presented in figures. Response 14: We thank the reviewer for this constructive suggestion. In response, Tables 2 and 3 have been re-organized and reformatted with the aim of improving both clarity and readability. The revised layout now integrates categorical interpretations (S, I, R) with MIC values in a more intuitive way, allowing an easier and more immediate interpretation of the results. With this new structure, we consider that the tables already provide a comprehensive and accessible overview, and therefore the addition of figures may not be necessary. We hope that this revised format can meet the concerns of the reviewer |
|
15: In the Materials and Methods, how did the authors identify the diverse species of the wild birds? Response 15: We thank the reviewer for this pertinent question. Most of the birds investigated originated from wildlife rescue centers (as described in the Materials and Methods, lines 594-599), which are staffed by biologists, ornithologists, and specialized veterinarians. As a result, the birds were already identified upon admission to the veterinary hospitals. Moreover, several bird species were common to both study sites and could be easily recognized. Nevertheless, to further clarify this aspect, precisely the moment in which birds’ identification has been done, a dedicated sentence has been added to the Materials and Methods section (lines 600-602). |
|
16: The experimental results lack statistical analysis, and doesn't specify the number of replicates or statistical tests used. Response 16: We sincerely thank the reviewer for this observation. We fully acknowledge the relevance of statistical analyses in strengthening experimental studies. However, in the present work, no statistical comparisons between the populations sampled at the two sites were feasible, due to the relatively limited sample size and the high degree of heterogeneity of the bird species included, both within each site and between the two sites. This limitation is explicitly discussed in the Discussion section (lines 576-578), where it is highlighted among the constraints of our study. In addition, the primary objective of this investigation was to provide a descriptive overview of the presence/absence of ESKAPE bacteria in wild birds and to assess their antimicrobial resistance profiles. With regard to replicates, these were not performed precisely because of the descriptive nature of the work, combined with the ethical and practical constraints of working with protected wild species. Furthermore, some individuals received pharmacological treatments during hospitalization, which would have introduced confounding variables and made repeated sampling of the same subjects inconsistent with the study’s objectives. Importantly, our goal was not to evaluate the acquisition of ESKAPE bacteria during hospitalization, but rather to provide a snapshot of their occurrence at admission. For all these reasons, the study was intentionally designed as a descriptive survey. Nonetheless, we are grateful for your observation, which highlighted that this aspect may not have been sufficiently clear in the original version of the manuscript. Following your suggestion, we have now added a dedicated paragraph in the Discussion section (lines 578-585) to better clarify this point. |
Reviewer 3 Report
Comments and Suggestions for Authors
Review
The growing antimicrobial resistance is a serious problem of great public importance. The acquisition of resistance mechanisms is more frequently observed among clinical isolates or those originating from anthropogenically affected habitats through selective pressure than among environmental ones. Wildlife is not expected to play a fundamental role in the emergence of antibiotic resistance due to the fact that it is not treated with antibiotics. However, wild animals can acquire antibiotic-resistant pathogens and act as their reservoir, transmitting them through horizontal transfer. Overall, antimicrobial sensitivity testing applied to wild-type strains is severely limited but very valuable approach. This study sheds light on interesting aspect of avian ecology. The article is well written, with only minor edits recommended in some paragraphs due to some technical remarks and stylistic imperfections.
Line 33: Enterobacter hormaechei not abbreviated because the species is mentioned for the first time
Lines 59-60: Due to unnecessary repetition of some words and expressions in the passage (environment, habitats, humans, domestic animals, wildlife), it is suggested to edit the sentence. For example: “Their persistence in different habitats and ability to engage in horizontal gene transfer (HGT) allow them to acquire resistance through interactions with other microorganisms [11]. In this respect, wildlife may serve as a link in the transmission cycle of AMR bacteria among humans, domestic animals, and the environment [12,13].”
Lines 78-79: Suggestion to remove this part of the sentence: “Considering that birds are emerging as potential reservoirs and sentinels for ARB, particularly ESKAPE bacteria [25]”, because it's too long, and essentially this is what was said above in lines 70-73
Lines 111-113: Suggestion: to indicate the number of isolates in numbers rather than words, because it is more clear and easy to understand. This also applies, for example, to lines 183, 205, 210, 257, 325, 388, 391
Line 161: The designations on the Figure 1a,b are illegible. Please try to provide a figure with clearer and more readable labels. Also, in figure captions (lines 163-165) the abbreviated form of species names can be given. Line 166: please delete in: “Distribution of and number of…”
Lines 222-224 and Lines 230-232: Suggestion to remove this sentence from the table captions: “MICs were interpreted using EUCAST, CLSI, CA-SFM, or ECOFFs (Supplementary Materials, Table S3); if no breakpoints were available, raw MIC values (µ g/mL) are shown”. This is already explained in the text, not need to repeat
Lines: 126, 195, 265, 302, 319, 341, 348, 384, 421, 438, 446: Abbreviation should be avoided at the beginning of the sentence
Line 418: Please delete in: “A special attention special should be…”
Line 448: SCCmec in italics
Please, unify everywhere in the text: Gram-… or gram-…
Author Response
|
Response to Reviewer 3 Comments
|
|
We would like to thank the reviewer for taking the time to review this manuscript and for the positive and encouraging comments. Please find below our detailed responses to each of your comments, with the corresponding revisions clearly highlighted in the revised manuscript using track changes. Due to the modifications and additions made to the text, the line numbers in the revised version differ from those in the original manuscript. The line references provided in our responses correspond to the revised file.
|
|
Comment and Suggestions |
|
Comment 1: Line 33: Enterobacter hormaechei not abbreviated because the species is mentioned for the first time.
|
|
Response 1: We agree with the reviewer’s comment. You will not find the correction applied to the previous version of the text, for which a complete rewrite was requested; however, we made sure to take your comment into careful consideration when preparing the revised version of the abstract. |
|
Comment 2: Lines 59-60: Due to unnecessary repetition of some words and expressions in the passage (environment, habitats, humans, domestic animals, wildlife), it is suggested to edit the sentence. For example: “Their persistence in different habitats and ability to engage in horizontal gene transfer (HGT) allow them to acquire resistance through interactions with other microorganisms [11]. In this respect, wildlife may serve as a link in the transmission cycle of AMR bacteria among humans, domestic animals, and the environment [12,13].”
|
|
Response 2: We sincerely thank the reviewer for the suggested reformulation of the sentence, which helped us to clarify and improve it. The text has been revised accordingly (lines 79-82). |
|
· Comment 3: Lines 78-79: Suggestion to remove this part of the sentence: “Considering that birds are emerging as potential reservoirs and sentinels for ARB, particularly ESKAPE bacteria [25]”, because it's too long, and essentially this is what was said above in lines 70-73. |
|
Response 3: As thoughtfully suggested, this part has been removed from the text to avoid redundancy (lines 119-120). |
|
· Comment 4: Lines 111-113: Suggestion: to indicate the number of isolates in numbers rather than words, because it is more clear and easy to understand. This also applies, for example, to lines 183, 205, 210, 257, 325, 388, 391.
Response 4: We thank the reviewer for this observation. Our initial choice to spell out numbers from one to ten was based on a general editorial convention. However, we have revised the text by replacing words with Arabic numerals in order to improve clarity and readability throughout the manuscript. |
|
Comment 5: Line 161: The designations on the Figure 1a,b are illegible. Please try to provide a figure with clearer and more readable labels. Also, in figure captions (lines 163-165) the abbreviated form of species names can be given. Line 166: please delete in: “Distribution of and number of…” · Response 5: We thank the reviewer for the observation. Figure 1 has been revised to improve the overall quality and the readability of the labels. In addition, the figure caption has been simplified: the abbreviated forms of species names are now used, and the redundant sentences has been removed (lines 230-233). In response to this comment, we considered it appropriate to also improve the resolution of Figure S1 in the Supplementary Materials. |
|
· Comment 6: Lines 222-224 and Lines 230-232: Suggestion to remove this sentence from the table captions: “MICs were interpreted using EUCAST, CLSI, CA-SFM, or ECOFFs (Supplementary Materials, Table S3); if no breakpoints were available, raw MIC values (µ g/mL) are shown”. This is already explained in the text, not need to repeat. Response 6: We agree with the reviewer. As suggested, the indicated part has been removed from the table captions (lines 326-327, 334-336). |
|
· Comment 7: Lines: 126, 195, 265, 302, 319, 341, 348, 384, 421, 438, 446: Abbreviation should be avoided at the beginning of the sentence
· Response 7: We thank the reviewer for the appropriate suggestion. The text has been revised accordingly (lines 183, 200, 211, 268, 371, 405, 410, 427, 450, 494, 532, 550, 558). |
|
Comment 8: Line 418: Please delete in: “A special attention special should be…” · · Response 8: We would like to thank the reviewer for pointed this out. We have changed it in the text (line 529). |
|
· Comment 9: Line 448: SCCmec in italics
Response 9: We agree with the reviewer. We have corrected it in the text (line 560). |
|
Comment 10: Please, unify everywhere in the text: Gram-… or gram-…
· Response 10: We thank the reviewer for this observation. The text has been revised accordingly, and all occurrences have been unified using “Gram-” with a capital “G” (lines 251, 274, 390, 537). |
Round 2
Reviewer 2 Report
Comments and Suggestions for Authors
Reviewer 2’s Comments and Suggestions for Authors
(Second Round)
Journal: Antibiotics, MDPI
Manuscript ID: antibiotics-3876574-R1
Type: Article
Title: Looking For ESKAPE Bacteria: Occurrence And Phenotypic Antimicrobial Resistance Profiles In Wild Birds From Northern And Central Italy Sites
Authors: Guido Grilli*, Maria Cristina Rapi, Laura Musa, Giacomo Di Giacinto, Fabrizio Passamonti, Stefano Raimondi, Oriana Cianca , Maria Pia Franciosini
The authors of the Manuscript ID: antibiotics-3876574-R1 have addressed most of my comments and suggestions. However, this manuscript should be further improved.
Essential revisions
1. In the “Response to Reviewer 2 Comments”, some line numbers of the revisions are not consistent with those shown in the manuscript, e.g., “the Discussion section (lines 578-585)”, “Materials and Methods section (lines 600-602)”, and “the Results section (lines 221-226)”.
2. In the Abstract: please delete the words “Background/Objectives:”, “Methods:”, “Results:” and “Conclusions:”.
3. Table 1: the authors should re-format this table to be more easily read.
4. What was the subtitle of the 2.5 section?
5. Figure 1: please write bacterial Latin names in italics.
6. Lines 89-110: please check the front size, and the subtitle “ Results”.
7. Please check the page layout of pages 8 to 13.
Since it is not easy to guide the authors to format Tables such as Table 1 by the Reviewer’s Report, I strongly suggest Editors in the Production Team of the journal Antibiotics to help the authors to format the Table 1 and Tables in Supplementary files.